# A Review of Bio-Based Activated Carbon Properties Produced from Different Activating Chemicals during Chemicals Activation Process on Biomass and Its Potential for Malaysia

**DOI:** 10.3390/ma16237365

**Published:** 2023-11-27

**Authors:** Tung Woey Chew, Paik San H’Ng, Bin Chuah Teong Guan Luqman Chuah Abdullah, Kit Ling Chin, Chuan Li Lee, Bin Mohd Sahfani Mohd Nor Hafizuddin, Lulu TaungMai

**Affiliations:** 1Institute of Tropical Forestry and Forest Products, Universiti Putra Malaysia, Serdang 43400, Malaysia; c_kitling@upm.edu.my (K.L.C.); chuanli@upm.edu.my (C.L.L.); hafizsahfani@gmail.com (B.M.S.M.N.H.); lulutawngmai@gmail.com (L.T.); 2Faculty of Forestry and Environmental, Universiti Putra Malaysia, Serdang 43400, Malaysia; 3Faculty of Engineering, Universiti Putra Malaysia, Serdang 43400, Malaysia; chuah@upm.edu.my

**Keywords:** activated carbon, chemical activation, biomass, chemical recovery

## Abstract

Activated carbon is the preferred adsorbent for gas and water treatment in various industry across the world due to its efficiency, reliability, and accessibility. Recently, in Malaysia, studies are mainly focused on the fabrication of activated carbon from lignocellulosic biomass-based precursors from agricultural waste such as coconut shell, rice husk, and palm kernel shell. Activated carbon fabrication is a two-step process; the precursor will first undergo carbonization, then, activation is carried out either physically or chemically to develop its porous surface for adsorption purposes. The main benefit of activated carbon is the customizable pore structure for different utilization, which can be easily achieved by the chemical activation process. The types and concentration of chemicals used for activation, pre-treatment of precursor, duration of the activation process, and the mass ratio of precursor to chemicals are proven to effectively influence the resulting pore structure. However, the chemicals used in the activation process can be harmful to the environment. Thus, the chemical recovery process is necessary after the activation process. Nonetheless, more in-depth research on producing activated carbon from abundant biomass materials with bio-based chemical agents for activation is needed to achieve an ecological and sustainable manufacturing process.

## 1. Introduction

Activated carbon (AC) is a kind of carbon that has undergone an activation process to make it exceptionally porous and capable of efficiently adsorbing different chemicals [1,2,3,4,5]. The mechanism of adsorption of AC is commonly due to the micropores present in the carbon or the weak Van der Waals forces that can attract the impurities. Activated carbon that is available in the market is made using carbonization and activation techniques on organic materials of plant-based origin [6,7,8]. Lignocellulosic biomass and coal materials have been used as the main raw materials for manufacturing of activated carbon [9,10,11]. The activation process is accomplished by heating and steaming the coal and biomass, which creates a structure with minuscule holes and fissures in the precursor material that may catch and retain particles from gases or liquids traveling through it [1,3,10]. Due to its enormous microporous structure, activated carbon is frequently utilized in a variety of industrial treatments, including chemical treatment, protective suits, water and air filtration, and environmental remediation [3,11,12,13,14].

Activated carbon is a popular adsorbent with several uses in numerous industries worldwide. Malaysia was the 16th leading exporter of activated carbon globally in 2021 with USD 51.3 million in exports. Activated carbon was Malaysia’s 360th most exported item that year. The top five countries to which Malaysia exports activated carbon are Thailand (USD 3.7M), the Netherlands (USD 7.62M), Japan (USD 10.6M), Singapore (USD 6.96M), and Chinese Taipei (USD 3M) [15]. It is frequently utilized in water and air purifying, food and beverage quality control, and medical purposes [16]. In recent years, activated carbon has been used for medical purposes, especially during the COVID-19 pandemic that emerged in 2020 [17,18].

In water treatment plants, activated carbon is commonly used to filter out contaminants such organic compounds, chemicals, and heavy metals [19,20,21,22]. Heavy metals and anions present in drinking water become a challenging public issue due to their causes in human health. Jaya Rajan and Indira Anish [13] pointed out that a removal efficiency of 150 mg/g was observed for cadmium removal using biomass-activated carbon. Aziz et al. [23] found out that the activated carbon produced in his study was efficient at adsorbing contaminants from water, with removal efficiencies up to 100%.

Furthermore, volatile organic compounds (VOCs) and other air pollutants can be captured using activated carbon, when it is designed in air quality monitoring systems. According to research by Das et al. [24], the ability of activated carbon to absorb the VOCs is dependent on the form of activated carbon. The activated carbon in the form of fiber was shown to have a higher capacity for adsorbing volatile organic compounds than the majority of other adsorbents that are available as pellets or powders, including granular activated carbon, zeolites, and silica gel.

Activated carbon is also extensively used in the food and beverage cultivation and processing to eliminate contaminants and enhance the quality of food products. Solís-Fuentes et al. [25] reported that activated carbon is efficient at removing pollutants from sugarcane juice in the cane sugar manufacturing process, with efficiency up to 98% at ambient temperature.

Moreover, activated carbon also has a long history of being employed in medical procedures including hemodialysis and the treatment of intoxication. Research by Mostafalou and Mohammadi [26] acknowledged that activated carbon was successful in eliminating poisons from circulation, and it was efficient in curing drug overdose and intoxication. Drugs and toxins can bind to activated carbon and this in return helps rid the body of unwanted substances.

There is research published in 2019 that demonstrated how effective it was in eliminating colors and other impurities from wastewater used in the dyeing process [27]. Maguana et al. [28] studied the adsorption process of activated carbon in dyes removal using the Dubinin–Radushkevich isotherm and revealed that the adsorption of methyl orange onto activated carbon was a physisorption process in nature. The adsorption capacity of activated carbon was found to be 336.12 mg/g at a temperature of 20 °C. Indonesia is a successful real-life example of a country that has incorporated activated carbon in various industries, particularly iodine production, water filtration systems, and the textile industry. This proves the efficiency and significance of activated carbon in numerous types of applications. The negative ∆G value of −25 kJ/mol obtained from the Langmuir equation in an iodine recovery study by Jha and Jha [29] indicated that the adsorption process was a spontaneous, feasible, and the physio-chemical type. After iodine has been obtained from seaweed during the iodine manufacturing process, impurities and pollutants are removed from the iodine using activated carbon. The iodine solution is put through a sheet of activated carbon during the process, which draws out the contaminants and leaves behind pure, high-quality iodine [30].

### 1.1. History of Activated Carbon

From ancient times, activated carbon has been used to purify materials, and it was first utilized medicinally in the 18th century. Johann Lowitz, a Russian scientist, made the original discovery of charcoal’s liquid decolorizing abilities in 1776 while looking for substances able to purify tartaric acid solution from brown, oily phlogiston [31]. In the early 1800s, the process for activating carbon was officially discovered. In order to make a porous, adsorbent substance that could remove pollutants from liquids and gases, it was initially made in the early 20th century by burning coal, wood, or coconut shells. It was utilized in the 1940s for different purifying and extraction procedures in the chemical and food industries, as well as in gas masks and protective equipment during World War II. Since then, activated carbon has been omnipresent in a variety of uses, including pharmaceutical, medicinal, and environmental monitoring [5,32].

### 1.2. Activated Carbon from Coal

Activated carbon can be made from a variety of materials, notably coal, coconut shells, wood, and lignite. Due to its accessibility and affordability as compared to other resources, activated carbon made from coal has become one of the most often utilized forms. The preparation process consists of phosphoric acid impregnation followed by carbonization in nitrogen at 400–600 °C for 1–3 h. Alternatively, high-quality coal is heated to a temperature between 900 °C and 1100 °C in the absence of oxygen to produce coal-based activated carbon, which eliminates most volatile materials and leaves behind a microporous, high adsorption substance [33].

Coal-based activated carbon used to be favored by industrial users compared to other sources of activated carbon because of its outstanding ability to adsorb and stability in a variety of acidic and basic conditions [34,35]. In addition to that, coal is a by-product of the coal mining process and is easily accessible in many areas. Moreover, there is flexibility in the source of materials since coal-based activated carbon may be produced from both bituminous and sub-bituminous coal.

Wang et al. [34] tested several coal-based activated carbons in eliminating pollutants from waste water, and the results are very promising, with nearly 90% of the pollutants being successfully removed. Further, in his study, lignite coal-derived activated carbon showed the best capability for adsorbing both organic and inorganic contaminants, which makes it an efficient adsorbent for treating wastewater.

Zheng et al. [35] noted that the process of carbonization and the activation process on the coal materials significantly changed the molecular structure of raw coal, and a large number of organic functional groups were diminished during the process. Nonetheless, the carbonization process employed on the coal enriched the pore structure of coal by thermal ablation, and it has a pore expansion effect on all the pores in coal, while the activation process created micropores in the raw coal.

### 1.3. Activated Carbon from Biomass

Recently, the focuses of activated carbon production have been shifted from coal to biomass as raw materials. The accessibility of biomass, low cost, and sustainability of biomass have made it a popular precursor for the manufacture of activated carbon in recent years. Because of their ready availability and sustainability, biomass materials including coconut shells, wood, and bamboo are frequently used to produce activated carbon. Activated carbon produced from biomass has a number of advantages over the conventional material (coal), including environmental sustainability, economic efficiency, and the use of renewable resources.

In recent research by Rostamian et al. [36], rice husk was chemically activated using potassium hydroxide to create activated carbon and a large surface area (2201 m^2^/g) covered with a total pore volume of 0.96 cm^3^/g with an excellent sodium adsorption capability with a capacity of 134.2 mg/g was recorded. Khuong et al. [37] used physical activation with CO^2^ to create activated carbon using a by-product from bamboo hydrothermal treatment. The resultant substance was well suited for use in carbon capture due to its large specific surface area of 2132 m^2^/g and high capacity for carbon dioxide adsorption.

Many researchers discovered that activated carbon has a sizable capacity for adsorption and would work well as an adsorbent to remove organic contaminants. This can be executed by manipulating the type of biomass for the AC precursor, the type of activation process, and the condition of the activation process to fabricate the total surface area and total pore volume of the AC required [12,38,39,40].

Moreover, Maniarasu et al. [41] were able to find a simple, affordable, and scalable approach for producing activated carbon from agricultural wastes including coconut shell and palm kernel shell. The findings demonstrated that the activated carbon made from agricultural waste have superior adsorption capabilities and a large surface area. The study came to the conclusion that biomass-derived activated carbon might be a useful tool for producing high-quality adsorbents and managing trash in a sustainable manner.

## 2. Biomass as a Precursor in AC Production

Two prevalent precursors for the creation of activated carbon are biomass and coal. While coal has historically been used to make activated carbon, biomass is now gaining popularity because it is abundant and renewable [42].

Due to its inexhaustible nature, activated carbon derived from biomass is thought to be a more ecologically responsible choice than activated carbon derived from coal [43,44]. Furthermore, by using this material, abundant waste can be re-valued and marketed at a better price. Additionally, using abundant biomass as precursor will also help in waste management issues globally, as waste is turning into a useful product to eliminate pollution in air and water [9].

Yong et al. [45] found out that the environmental impact of biomass-based AC is significantly lower than coal-based AC when both ACs are evaluated using the life cycle assessment (LCA) framework outlined in ISO 14040. Additionally, it has been discovered that activated carbon made from biomass has superior adsorption capabilities compared to activated carbon made from coal as shown in Table 1 [46,47].

Biomass is favored for AC production over the other raw material; coal is a finite resource, its mining and the processing of coal have a detrimental effect on the ecosystem, such as causing air and river pollution [51]. According to Bian et al. and Goswami [52,53], coal mining and processing can have significant environmental impacts including air pollution, land subsidence, wetlands destruction, and mining dumps, as well as landscape alteration and environmental contaminations.

The interest in using biomass-based activated carbon as a sustainable alternative to coal-based activated carbon is due to its renewable and environmentally friendly properties [39,54]. Thus, despite the fact that coal-based activated carbon has long been a popular choice among industrial users, interest in biomass-based activated carbon as a more environmentally friendly substitute is rising.

### 2.1. Type of Biomass for AC Production

In general, activated carbon can be produced from any lignocellulosic material that is high in carbon content and low in organic and volatile strength [42]. Some selected lignocellulosic material can produce highly porous activated carbons with large surface areas. Lignocellulosic materials, referred to as biomass, usually go through the process of carbonation (some may refer to this as the pyrolysis process) followed by chemical or physical activation for the creation of activated carbon. The characteristics and effectiveness of the resulting activated carbon (AC) can be significantly influenced by the choice of organic matter (biomass) precursor.

The viability of various kinds of biomass raw materials for the creation of activated carbon has been researched [38,42,55]. These include timber leftovers such as saw dust and wood pieces as well as farming remnants including rice husks, sugarcane bagasse, and maize cob. As possible sources for the creation of activated carbon, different refuse materials, including urban solid waste, sewage, and wastewater, have been further investigated. The Table 2 below shows the types of biomass used for AC fabrication and their characteristics.

Agricultural waste is the most commonly used biomass source because of its high contents of cellulose and lignin percentages. AC produced from these sources were found to have excellent thermal stability and were shown to withstand many adsorption cycles [57,58].

Rice husks, a by-product of the milling of rice, are a potential precursor for the development of activated carbon due to their significant amount of carbon and low level of ash, according to research by [36]. According to the research, rice-husk-based activated carbon had a substantial surface area and strong sodium adsorption capabilities.

The use of sugarcane bagasse, a by-product of sugarcane refining, as a precursor for the development of activated carbon was the subject of another study by Amin and Mall et al. [56,59]. The research discovered that activated carbon made from sugarcane bagasse had a large surface area and effective capability for adsorbing Congo red and reactive orange dyes.

Danish and Ahmad and González-García [7,9] addressed the use of different biomass precursors for the creation of activated carbon in their review paper. They discovered that waste materials, timber leftovers, and farming wastes all had the potential to serve as precursors for the development of activated carbon. Their usefulness depended on variables including access, expense, and composition.

In general, a broad variety of substances derived from biomass have been examined for their potential as starting points for the development of activated carbon. Accessibility, expense, as well as the desired properties of the resulting activated carbon are a few examples of the variables that affect the choice of biomass precursor.

While many precursors can be used for biomass-derived AC synthesis, the choice is mainly based on the geographical location, biomass availability, cost, and the application for which they will be used.

### 2.2. Biomass in Malaysia for AC Production

Malaysia is a nation with numerous sources of biomass that are beneficial for producing activated carbon. About 168 million tons of biomass are produced annually in Malaysia, including lumber, palm oil waste, rice husks, coconut stem fibers, municipal waste, and sugarcane waste [60]. Palm kernel shell (PKS), coconut shell, sawdust, and rice husk are some of the biomass precursors that are most frequently used for the creation of activated carbon. The popularity of these materials is directly associated to Malaysia’s farming and timber industries; these precursors are readily available and easy to obtain there.

Among of all of the biomass mentioned, coconut shell is the most common precursor for the development of activated carbon in Malaysia. One of Malaysia’s major sectors is the coconut industry. In 2021, the output of coconuts is expected to be worth around MYR 666.70 (USD 158.70) million. Table 3 shows the types of coconut-based product that had been traded.

The export value of products made from coconut grew from over MYR 700.60 (USD 166.80) million in 2016 to more than MYR 1.14 (USD 0.27) billion in 2020 [61]. The Table 4 below shows coconut-shell-based products that had been exported to ASEAN countries from 2016 to 2020.

Coconut shell (CS) is a great source of activated carbon because it contains much lignin and is very permeable [62]. In Malaysia, CS is usually heated with the assistance of a burning agent, such as carbon dioxide or steam, and then physically activated to develop activated carbon [62]. Despite how AC made from coconut shell can achieve a high BET value up to more than 1600 m^2^/g, researchers are turning their focus to palm biomass as the coconut industry is downsizing due to low profit compare to other crops [61].

Palm biomass in this context refers to the palm kernel shell (PKS); being abundantly available with ease of accessibility make it the most suitable material for the development of activated carbon. This may be due to the fact that Malaysia was the world’s second-largest exporter of palm oil in 2021, supplying a value of USD 15 billion abroad [63]. PKS with a special chemical composition, in which it is extremely porous and contains much carbon, is a perfect precursor for the creation of activated carbon [64,65,66,67]. The chemical composition of PKS-derived AC poses the capability of removing hydrogen sulfide, cadmium, chromium, and lead in water, which makes it suitable for wastewater, and to a certain extent for palm oil mill effluent (POME) treatment. In Malaysia, a two-step method that involves carbonization to turn PKS into biochar through pyrolysis, then chemical or physical activation is carried out to activate the carbonaceous material into activated carbon equipped with an enormous pore volume [66,68].

Research by Andas et al. and Lee et al. [69,70] found that PKS-based activated carbon generated by two-step activation process had an extensive surface area and excellent methylene blue dye adsorption capability. The research also revealed that the properties of the activated carbon that was produced were greatly impacted by the activation procedure.

Lee et al. [49] published research related with the use of PKS as a biomass in the physical activation of carbon to produce activated carbon [49]. In the research, the final activated carbon had a large surface area and strong adhering capabilities for the color’s methylene blue and iodine after being post-treated with ultrasound.

The possible application of different kinds of biomass precursors, including PKS, for the production of activated carbon in Malaysia was covered by Andas et al., Lee et al., and Zainal et al. [67,69,70]. PKS, according to the researchers, is an appropriate precursor for the creation of activated carbon because of its high carbon concentration and minimal ash content.

Additionally, Lee et al. [70] examined the characteristics of activated carbon made from PKS and coconut shells in other research. According to the study, PKS is a potential precursor for the manufacture of activated carbon in Malaysia because it has a larger surface area and greater methylene blue adsorption capability.

PKS is a readily available abundant by-product of the palm oil industry and offers favorable characteristics for the manufacture of activated carbon, making it an ideal precursor for activated carbon in Malaysia.

Additionally, another biomass material that is commonly employed in Malaysia to produce activated carbon is sawdust. In Malaysia, sawdust is a readily available by-product of the wood industry. Sawdust is an excellent precursor for the development of activated carbon because of its high carbon concentration and high porosity [71]. In Malaysia, sawdust is usually chemically activated into activated carbon using a method that includes administering an activator, such as phosphoric acid, to the material [71].

Another biomass that can be utilized in Malaysia to produce activated carbon is rice husk. In Malaysia, rice husk, a residue of rice processing, is widely available. Due to its high silica concentration and high porosity, rice husk is an ideal resource for the development of activated carbon [72]. In Malaysia, rice husk is usually heated in the presence of a vaporizing substance, such as carbon dioxide or steam, and then physically activated to create activated carbon [72].

## 3. Activation Process for Biomass-Based Activated Carbon Production

Physical and chemical activation are usually the two primary methods used to produce activated carbon from biomass precursors. Physical activation entails thermally decomposing the biomass precursor in the absence of an oxidizing substance, then charring the resulting residue with a gas such as carbon dioxide or moisture to activate it. In order to achieve an extensive surface area and an established pore structure, the procedure normally is conducted under monitored conditions at temperatures varying from 600 °C to 1000 °C [2].

The process of physical activation has the benefit of producing activated carbon with a substantial surface area and superior adhesion capabilities. In comparison to chemical activation, it is also a simpler procedure and needs less capital to operate [2,68].

However, physical activation necessitates lengthier activation periods and higher temperatures, which increases the expenditure of energy. In comparison to chemical activation, physical activation produces activated carbon with a less homogenous particle structure [2,66].

In contrast, chemical activation entails impregnating the biomass precursor with a chemical agent, typically a basic or acidic substance, and then thermally decomposing the precursor under carefully regulated circumstances [1]. A structure with an enormous surface area and a clearly established pore diameter variation is created as a consequence of the procedure.

Chemical activation has the benefit of enabling a greater degree of modification of the pore structure of the activated carbon, improving adsorption characteristics. In addition to being less time-consuming and requiring lower temperatures than physical activation, the process is also simpler [49,73].

## 4. Chemical Activation Process

As mentioned in the previous section, one approach to manufacturing AC involves chemical activation processes where several agents are utilized to obtain the desired absorption properties. In this section, various chemical activation methods that are employed while producing biomass-derived AC with diverse chemicals as catalysts are discussed.

### 4.1. Chemicals Used for Activation

Activation of biomass through chemical means is commonly known as chemical activation. Using various chemicals such as potassium hydroxide (KOH), sulfuric acid (H_2_SO_4_), sodium hydroxide (NaOH), zinc chloride (ZnCl_2_), and phosphoric acid (H_3_PO_4_) leads to the development of porosity within the structure being reported [42,74]. In order to produce activated carbon with the desired properties, an impregnation phase is considered important, where the chemicals are absorbed by the precursor materials prior to high-temperature carbonization. This will eliminate all unwanted substances from the mixture [74]. The surface area of most activated carbon from lignocellulosic biomass by H_3_PO_4_ had a variability of 456.1–2806 m^2^/g, yielding 26.1–85%, and an extreme adsorption capacity of 2.5–89.29 mg/g [55].

The activation of biomass-based activated carbon often employs phosphoric acid (H_3_PO_4_) as an activation chemical agent. It is commonly utilized due to its capacity for generating high micropore volume, low ash content, and a high surface area [75]. Li et al. [72] stated that the use of H_3_PO_4_ in the activation process of rice husk can yield a surface area up to 1016 m^2^ g^−1^ when under the conditions of a solid material to H_3_PO_4_ mass ratio of 1: 2 and with an activation temperature of 500 °C for 1 h.

Additionally, KOH is also used as an activation chemical for biomass during chemical activation. KOH activation has the ability to produce mesopore volumes and high surface areas in the biomass according to Barelli et al. and Montes et al. [76,77]. Foo and Hameed [62] found out that NaOH produces higher levels of both micropores and surface area across biomass than most other chemical agents. KOH can react with active O-containing species in biomass at lower chemical to biomass ratios (1:8–1:2) and lower temperatures (400–600 °C). KOH will completely transform to K_2_CO_3_ after the activation, leading to the formation of large amounts of gaseous products and phenols (reaching 75%). With a significant decrease in the phenols and O-species, the hydrocarbons became the dominant species (reaching a content of 57.43%). O-containing groups further transformed into more stable OH, CO, and COOH groups. Both mesopores and micropores are formed as a result of the intercalation of potassium into the carbon network during the activation.

The chemical agent zinc chloride, abbreviated as ZnCl_2_, is utilized to activate biomass-based activated carbon. This activation method yields a high surface area and mesopore volume in the resultant activated carbon [78]. In zinc chloride activation, the liquid chemical is intercalated into the carbon matrix to produce pores at a temperature above the melting point of the chemical agent. The reaction between the carbon atoms and dehydrating agent is promoted in the extended interlayers of carbon. The application of ZnCl_2_ in chemical activation generally improves the carbon content through the formation of aromatic graphitic structure. ZnCl_2_ has a low melting point of 283–293 °C allowing better contact with the carbon surface above a 500 °C activation temperature [8,79].

Sulfuric acid (H_2_SO_4_) is also used for activating biomass-based activated carbons. Its efficacy stems from producing highly mesoporous materials that have large surface areas when converted into active forms due to the heteroatom doping effect induced by sulfur on the material’s structure during preparation [80].

Calcium chloride (CaCl_2_), potassium carbonate (K_2_CO_3_), and hydrochloric acid (HCl) are among the additional chemical agents utilized for the activation of biomass-based activated carbon. An article by Ahmad et al. [14] discovered that CaCl_2_ was an effective agent for producing activated carbon with high micropore volume and a substantial surface area, while K_2_CO_3_ is renowned for generating activated carbon with high mesopore volumes and large surface areas [62]. In contrast, HCl is used to produce activated carbon with a high surface area and high micropore volume [81,82].

The choice of chemical agents used in the biomass activation process is determined by the specific properties that are sought in producing an effective form of activated carbon that is suitable for various applications.

### 4.2. Factors Affecting Chemical Activation Process

#### 4.2.1. Concentration

As reported by Yang et al. [83], the increase in the concentration of KOH solution will lead to the increase in the adsorption value of iodine and methylene blue of the activated carbon, the highest yield of 24.48% at the concentration of 40% for iodine adsorption, while adsorption value for methylene blue is 1521.94 mg/g and 21.28 (10 m/g) at the concentration of 50%.

#### 4.2.2. Pre-Treatment

The pre-treatment effect of acid from three different activation agents, HCl, HNO_3_, and H_3_PO_4_, on the properties of activated carbons does not seem to significantly influence the final activated carbons. The values of specific surface areas and pore volumes showed a change from 506 m^2^/g to 616 m^2^/g and from 0.23 cm^3^/g to 0.27 cm^3^/g as reported by Bergna et al. [84] when HCl and HNO_3_, were used. However, Zhu et al. [85] mentioned that phosphorous acid pre-treatment during the KOH chemical activation process proved to be an effective strategy to prepare highly porous AC from sawdust, which allowed for maximum adsorption capacities of 303.03 mg/g at 30 °C. These data imply that the adsorption was an endothermic process.

#### 4.2.3. Duration of Activation

Ceyhan et al. [86] stated that the reaction time during chemical activation affects the iodine number of activated carbon. In his research, the iodine number values increased gradually from 564 to 1004 mg/g while increasing the activation time from 1 to 3 h. The low temperature employed in his research indicates that the increases in the iodine number values with activation time indicate that prolonged heat treatment is required for the full development of porosity at 300 °C. After an activation time of 3 h, the iodine number of the activated carbon decreased from 1004 to 680 mg/g, indicating that the longer duration of activation time caused some of the pores to enlarge and collapse.

In another study, the effects of different activation times during the chemical activation process on the properties of activated coke were examined by Ahmed and Theydan [87]. The activated coke was obtained from the jujube nucleus with zinc chloride with activation times from 0.5 h to 3.5 h. The study pointed out that the coke yield decreased as activation time increased from 0.5 to 3.5 h. This may be due to the fact that the volume of mesopores increased during the first 1.25 h. With the increase in activation time, mesopores began to collapse into larger pores.

#### 4.2.4. Mass Ratio

Zakaria et al. [88] elicited that the highest activated carbon yield of nearly 50% was obtained for the process of chemical activation using H₃PO₄ on mangrove timber prepared at an impregnation ratio of material to chemical of 3:1 and at an activation temperature of 300 °C. At the same time, the activated carbon prepared at an impregnation ratio of 4:1 at the same temperature of 300 °C resulted in activated carbon with the highest adsorption capacity of 72.3 mg/g at an initial concentration of 150 mg/L.

### 4.3. Chemicals Recovery after Chemical Activation of Biomass for Activated Carbon

The process to produce activated carbon from biomass using chemical activation results in biomass with high surface areas and pore volumes. Nonetheless, the main problems with this process are the large volumes of chemicals required and the waste water created. For example, KOH is poisonous and toxic in nature. KOH has a health hazard rating of 3, which creates adverse effects in the environment, especially a water body, and it promotes eco-toxicity in water.

When the chemical agent KOH is used during the activation process, in addition to the activated carbon (AC) with a high specific surface area, this process generates dissolved alkali metals, existing mainly as K_2_CO_3_. Some authors suggest the possibility of recycling the chemical; however, K_2_CO_3_, which is transformed from KOH after activation, is normally a less effective activation agent. Montes and Hill [77] used different KOH/K_2_CO_3_ ratios in the recovered solution as an activation chemical and they concluded that the higher the fraction of the more active form of potassium (i.e., KOH), the higher the level of activation in subsequent experiments. In another study, activated carbon from grape seed with the highest surface area of nearly 1300 m^2^g^−1^ was obtained at 800 °C in 50 wt% concentrated K_2_CO_3_ [73]. Foo and Hameed [89] depicted that microwave heating for K_2_CO_3_ chemical activation on rice husks created high surface area activated carbons with a BET surface area, total pore volume, and monolayer adsorption capacity of 1165 m^2^/g, 0.78 cm^3^/g, and 441.52 mg/g, respectively.

Wu et al. [90] described how biomass-activated carbon production using ZnCl_2_ can simultaneously recover the activator at a special temperature and pressure range. In the process, over 99.99% of the zinc used for activation can be recovered in the form of zinc chloride and metallic zinc. Pua and Zaini [79] stated that the residual activator of ZnCl_2_ on activated carbon can be recovered using Soxhlet unit, and reused for subsequent activations. The washing step that drains off the chemical will create serious pollution in the environment because of the hazardous nature of ZnCl_2_.

## 5. Future Research on the Activation Process for Biomass

The initiation method is an integral phase in generating activated carbon from biomass. Activated carbon boasts multiple uses in remedying environmental issues, purifying water and air, as well as storing energy. In contrast to coal-based activation techniques that carry adverse ecological effects through extraction and processing, the use of bio-sourced activated carbon has gained recognition due to its sustainable nature. Nevertheless, producing consistent quality activated carbons could become problematic owing to varying properties present across different kinds of biomass materials. This requires a solution for efficiency improvement on production sustainability via extensive research conducted towards this end objective regarding future studies surrounding activating processes for various types of biomasses involved with said challenges.

An area for future investigations is the enhancement of activation circumstances applicable to distinct types of biomasses. Activating necessitates engaging a physical or chemical component that generates a porous structure within the biomass, intensifying its surface expanse and adsorption potentiality. The conditions involved in activating, such as temperature, duration period, and variant agents, can significantly impact both porosity magnitude and surface chemistry with respect to activated coal. Consequently, researchers have room for deliberation into optimal situations when dealing with diverse kinds of biomass. These situations are explored to produce an alluring outcome with their desired properties concerning activated carbon acquisition opportunities by studying various existing studies carried out over time [91,92].

In the future, research may entail developing environmentally friendly activation agents that can be sustainable. The current methods including chemical reagents or steam contribute negatively to our ecosystem, generating harmful waste substances that need proper disposal protocols. In light of this development, experts could consider alternative means such as bio-based chemicals whose composition is replenishable with no risk at all for environmental harm caused by their usage alongside hazardous wastes from common actuation mediums employed currently. For instance, recent studies have discovered promising results utilizing some organic compounds such as citric acid and tartaric acid extracted through biomass sources rather than synthetic processes; these comprehensive efforts offer sustainable ways to generate activated carbon materials essential for filtration procedures [93,94].

Moreover, forthcoming research can delve into the prospect of amalgamating various activation methodologies to create activated carbon with improved attributes. One possibility entails integrating physical and chemical activation techniques that have shown promise in generating a high surface area and microporosity, as well as mesoporosity [66,95,96]. In addition, by utilizing multiple methods for activation purposes it is feasible to decrease energy consumption while increasing efficiency concerning the time required for said process.

Furthermore, carbon-based products can be used as solar energy absorbents (solar light receivers) with high light-to-heat conversion efficiency. This is primarily due to a wide spectrum of solar absorption, the ability to achieve low bulk density (high porosity), and the capacity to float on the surface of water.

To summarize, forthcoming investigations on the process of activating biomass can tackle the issues linked with manufacturing activated carbon that is consistent in quality and enhance its sustainability as well as efficiency. Some scientists have prospects to refine activation circumstances for different categories of biomass, create new sustainable agents for activation that are eco-friendly, and explore how multiple techniques could amalgamate into one system. These endeavors may contribute towards establishing an industry producing ecologically sound activated carbon that stays viable over time.

## 6. Summary

Activated carbon has been used for centuries for its adsorption ability. In recent years, there has been a growing preference for biomass-based precursors, as they are renewable and sustainable alternatives to coal-based activated carbon. Biomass-based activated carbon has a number of advantages over other types of activated carbon, including its high surface area, porosity, and adsorption capacity. These properties make it ideal for a wide range of applications, such as water purification, gas separation, and catalysis.

In addition to its use in activated carbon production, biomass can also be used to recover chemicals after chemical activation. This can help to reduce the environmental impact of activated carbon production, as the chemicals would otherwise be released into the environment. However, more research is needed in this area, as there is currently not enough information on how to recover chemicals effectively.

Overall, activated carbon is a versatile material with a wide range of applications. Biomass-based activated carbon is a promising new option that is more sustainable than coal-based activated carbon. With further research, biomass-based activated carbon could become a more widely used material in a variety of industries.

## Figures and Tables

**Table 1 materials-16-07365-t001:** BET values and micropore volumes of different precursors for activated carbon.

Precursor	BET Value (m^2^/g)	Micropore Volume (cm^3^/g)	References
Coal-based			
Solumnar coal	334.27	0.93	[48]
CBAC	165.815	0.006	[34]
Biomass-based			
Coconut shell	1687	0.79	[49]
Sugarcane bagasse	1113	0.5	[50]
Apricot stones	1000	n.a	[46]
Palm kernel shell	465	0.63	[49]

**Table 2 materials-16-07365-t002:** Types of biomass used for AC fabrication and their characteristics.

Biomass Type	Characteristics	References
Rice Husk	strong sodium adsorption capabilities 134.2 mg/g)	[36]
Sugarcane Bagasse Pith	capability for adsorbing Congo red and reactive orange dyes	[56]
Wood Waste	can separate a wide range of organic and inorganic pollutants	[7]

**Table 3 materials-16-07365-t003:** Types of products for exportation in coconut industry.

Kernel-Based	Fiber-Based	Shell-Based
- Desiccated coconut	- Coconut fiber	- Activated carbon
- Powdered coconut	- Coconut fiber products	- Charcoal
- Coconut milk		
- Copra		
- Copra meal		
- Coconut oil		
- Fresh coconut		
- Young coconut		

**Table 4 materials-16-07365-t004:** Export of coconut-shell-based products, (MYR).

Countries	2016	2017	2018	2019	2020
China	6,777,949	10,220,273	12,022,593	16,194,192	8,367,683
Japan	0	32,101,774	44,771,065	58,632,491	36,040,141
Indonesia	1,246,959	237,580	3,329,247	3,895,019	2,239,971
Philippines	935,938	1,174,205	1,459,667	1,207,539	1,463,244
Singapore	1,928,691	1,926,385	2,366,837	2,789,879	2,305,071
Thailand	5,207,938	4,677,472	4,046,799	4,358,462	4,798,868
Vietnam	4,207,343	5,532,209	5,814,425	5,875,996	4,007,583
Other countries	33,236,910	38,164,625	37,625,715	48,625,590	47,923,918
**TOTAL**	**86,518,473**	**99,492,247**	**111,436,348**	**141,577,168**	**107,146,479**

Source: METS Online DOSM, 2020. Note: RM1.00 = US$ 0.42.

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
