# Peer review of "A Review of Bio-Based Activated Carbon Properties Produced from Different Activating Chemicals during Chemicals Activation Process on Biomass and Its Potential for Malaysia"

_materials, 2023, doi:10.3390/ma16237365_

Round 1

Reviewer 1 Report

The authors have reported a review in the field of fabrication of biomass-based activated carbon. The focus of the review is the process of chemical activation, with the recovery of the used chemicals. The manuscript is well-written, but it doesn’t contain any figures or tables, which makes the presented data less straightforward to follow. Based on the number of publications on the topic of biomass-based activated carbon in recent years, the number of references used for writing this review is insufficient. There are more issues concerning the present paper.

The title of the manuscript should be changed, as it is unclear which process parameter are the authors referring to. Further, the section concerning chemical recovery (line 397) is only 25 lines long and contains only 5 references, which is insufficient to be a focus of the review paper.

In the abstract and one subsection title (line 217) of the manuscript, the authors mention biomass originating specifically from Malaysia. It is unclear whether the paper presents only the activated carbon fabricated from biomass originating from Malaysia. If this is the case, this should the noted in the title of the manuscript, otherwise authors should elaborate the reason why focusing on this specific biomass origin. There are only around 12 references concerning Malaysian-originating biomass. The authors are referring to a review paper (line 246), while none of the references 64, 66, and 67 are a review.

In subsection 1.2 (line 99) authors briefly present the fabrication of activated carbon from coal. As noted in line 114, coal-based activated carbon has good adsorption characteristics, while it is noted that biomass-based activated carbon is superior to coal-based (line 164). The authors should provide a table of data from the references comparing the adsorption capacities, specific surface areas, or other characteristics of coal-based with biomass-based activated carbons.

Smaller issues:

line 71: It is unnecessary to write the name of the journal from the reference. Consult the guide for authors.

line 393: It is more common to write ratios as 3:1, and 4:1.

line 305 and other places: Incorrect writing of chemical formulas.

line 345: A reader of this paper would probably already be familiar with the formula of sulfuric acid.

Reviewer 2 Report

Dear Authors,

Chemical Activation Process Parameter and Its Chemicals Recovery for Biomass-based Activated Carbon: A Review has been reviewed and the following observations have been made:

1.      This review lacks figures and tables, which can summarize other research findings.

2.      Authors can explain the adsorption mechanism in context with biomass-based activated carbon.

3.      Authors can prepare a table containing the title, scope, and references for the central review focussing on the topic of the role of biobased activated carbon in chemical recovery.

You can take the reference of these papers  

Kumar, A., Kumar, V. A Comprehensive Review on Application of Lignocellulose Derived Nanomaterial in Heavy Metals Removal from Wastewater. Chemistry Africa 6, 39–78 (2023). https://doi.org/10.1007/s42250-022-00367-8

Sharma, R., Jasrotia, K., Singh, N. et al. A Comprehensive Review on Hydrothermal Carbonization of Biomass and its Applications. Chemistry Africa 3, 1–19 (2020). https://doi.org/10.1007/s42250-019-00098-3

4.      There is no chemistry in this manuscript,

5.      It is advised to discuss the suitable models related to adsorption kinetics.

6.      Please check the typographical errors present in the manuscript.

7.      Abstract and summary need to be modified. Please do not use abbreviations in the abstract section.

8.      Most of the sentences used in the manuscript are vague. Please check the grammar also.

9.      Improvement is required in the   ‘’Future Research on Activation Process for Biomass’’ section.

10.   Microwave-assisted activation process can be included in the present review.

The quality of English language needs to be improved. 

Reviewer 3 Report

The manuscript provides an overview of the latest advances in the manufacture of activated carbon from lignocellulosic biomass precursors from agro-cultural waste. The manuscript is not particularly informative and well organised to provide an in-depth discussion on this topic. Revisions are therefore required before it can be accepted on Materials. 

1. The abstract part should be a high summary of the whole content. What is the focus of this review?

2. Other cases can be added in the introduction, such as the use of hydrothermal carbonisation of orange peels 10.3390/pharmaceutics14102249; or https://doi.org/10.1016/j.biortech.2015.02.035 and so on.

3.   In my opinion, a paragraph on the "History of Activated Carbon" is unnecessary.

4. There are no representative images. Please add.

5. In addition to the pictures, the authors should also add tables showing the main works on this topic, to simplify reading.

6. Correct the numerous typos in the manuscript. For example in section 4.1 use subscript for all formulae.

Moderate editing of English language required

Round 2

Reviewer 1 Report

The authors have improved the manuscript to a certain extent, but issues from the original submission are still present.

The new title of the manuscript should be rewritten, as there seems to be an error.

The major issue with the present paper is that there are only a dozen references used in the manuscript about the use of biomass-based activated carbon from Malaysia. To improve the manuscript, the authors should consider adding more references about the use of biomass-based activated carbon from Malaysia.

If the focus of the review is the activated carbon produced from biomass originating from Malaysia, the authors should provide a more detailed introduction section that includes additional information on biomass characteristics specific to Malaysia, and information such as assessments of the annual production of biomass in the region, etc.

The manuscript would benefit from the inclusion of more graphical data, such as tables or figures, to help illustrate presented data.  This will help to make the content more engaging and accessible to readers. For example, the types of biomasses and their characteristics from section 2.1. could be presented as a table.

The writing of units is not uniform throughout the manuscript. 

Author Response

Dear Dr/Prof,

Thank you for the review of our manuscript.

Manuscript Ref. No.:  Materials-2539305

Title: A Review of Bio-based Activated Carbon Properties Produced from Different Activating Chemicals During Chemical Activation Process on Biomass and Its Potential for Malaysia Biomass

The comments were highly insightful and enabled us to improve the quality of our manuscript.  We have addressed the comments and amended the manuscript accordingly, especially the total number of references used to prepare this manuscript and the detailed corrections are as follow:

Reviewer 1

The authors have improved the manuscript to a certain extent, but issues from the original submission are still present.

The new title of the manuscript should be rewritten, as there seems to be an error.
The title had been changed to “A Review of Bio-based Activated Carbon Properties Produced from Different Activating Chemicals During Chemical Activation Process on Biomass and Its Potential for Malaysia Biomass”

The major issue with the present paper is that there are only a dozen references used in the manuscript about the use of biomass-based activated carbon from Malaysia. To improve the manuscript, the authors should consider adding more references about the use of biomass-based activated carbon from Malaysia.

If the focus of the review is the activated carbon produced from biomass originating from Malaysia, the authors should provide a more detailed introduction section that includes additional information on biomass characteristics specific to Malaysia, and information such as assessments of the annual production of biomass in the region, etc.
More references, details, figures and table on the biomass production in Malaysia had been added in Chapter 1 and 2.

The manuscript would benefit from the inclusion of more graphical data, such as tables or figures, to help illustrate presented data.  This will help to make the content more engaging and accessible to readers. For example, the types of biomasses and their characteristics from section 2.1. could be presented as a table.

 A few table had been added including table for section 2.1.

The writing of units is not uniform throughout the manuscript. 

This manuscript had been checked through Grammarly after some amendment.

Hope this current manuscript meet your standard.

Reviewer 2 Report

Dear Authors,

The manuscript has been improved, hence it can be accepted for publication.

Author Response

Dear reviewer, 

Thank you for reviewing this paper. I am deeply grateful for your time and effort in reviewing my work and appreciate your constructive and insightful comments, which have helped me to improve my manuscript. 

Your feedback has given me a lot to think about, and I am confident that my manuscript will be stronger as a result.

Thank you again for your help. I am grateful for your willingness to contribute to the peer review process. 

Reviewer 3 Report

I apologize for the persistent unsatisfaction with the revised version of the paper. It appears that significant flaws persist, including a lack of figures, insufficient discussion, a revision of the English, and the presence of repetitive and unnecessary statements. The text may be accepted after these revisions.

 Moderate editing of English language required

Author Response

Dear Dr/Prof,

Thank you for the review of our manuscript.

Manuscript Ref. No.:  Materials-2539305

Title: A Review of Bio-based Activated Carbon Properties Produced from Different Activating Chemicals During Chemical Activation Process on Biomass and Its Potential for Malaysia Biomass

The comments were highly insightful and enabled us to improve the quality of our manuscript.  We have addressed the comments and amended the manuscript accordingly, especially the total number of references used to prepare this manuscript and the detailed corrections are as follow:

Reviewer 3

I apologize for the persistent unsatisfaction with the revised version of the paper. It appears that significant flaws persist, including a lack of figures, insufficient discussion, a revision of the English, and the presence of repetitive and unnecessary statements. The text may be accepted after these revisions.

I apologize for the dissatisfaction on my work. Title had been changed. Some data with figures and tables had been added and amendment had been done for language error. Grammar check had been done by using Grammarly upon submission. Hope the current manuscripts meet your standard.